# Dynamics of Neutralizing Antibody Responses Following Natural SARS-CoV-2 Infection and Correlation with Commercial Serologic Tests. A Reappraisal and Indirect Comparison with Vaccinated Subjects

**DOI:** 10.3390/v13112329

**Published:** 2021-11-22

**Authors:** Constant Gillot, Julien Favresse, Vincent Maloteau, Jean-Michel Dogné, Jonathan Douxfils

**Affiliations:** 1Department of Pharmacy, Namur Research Institute for Life Sciences, University of Namur, 5000 Namur, Belgium; constant.gillot@unamur.be (C.G.); julien.favresse@slbo.be (J.F.); vincent.maloteau@unamur.be (V.M.); jean-michel.dogne@unamur.be (J.-M.D.); 2Department of Laboratory Medicine, Clinique St-Luc Bouge, 5000 Namur, Belgium; 3Qualiblood s.a., 5000 Namur, Belgium

**Keywords:** COVID-19, SARS-CoV-2, neutralizing antibody

## Abstract

Neutralising antibodies (NAbs) represent the real source of protection against SARS-CoV-2 infections by preventing the virus from entering target cells. The gold standard in the detection of these antibodies is the plaque reduction neutralization test (PRNT). As these experiments must be done in a very secure environment, other techniques based on pseudoviruses: pseudovirus neutralization test (pVNT) or surrogate virus neutralization test (sVNT) have been developed. Binding assays, on the other hand, measure total antibodies or IgG, IgM, and IgA directed against one epitope of the SARS-CoV-2, independently of their neutralizing capacity. The aim of this study is to compare the performance of six commercial binding assays to the pVNT and sVNT. In this study, we used blood samples from a cohort of 62 RT-PCR confirmed COVID-19 patients. Based on the results of the neutralizing assays, adapted cut-offs for the binding assays were calculated. The use of these adapted cut-offs does not permit to improve the accuracy of the serological assays and we did not find an adapted cut-off able to improve the capacity of these tests to detect NAbs. For a part of the population, a longitudinal follow-up with at least two samples for the same patient was performed. From day 14 to day 291, more than 75% of the samples were positive for NAbs (*n* = 87/110, 79.1%). Interestingly, 6 months post symptoms onset, the majority of the samples (*N* = 44/52, 84.6%) were still positive for NAbs. This is in sharp contrast with the results we obtained 6 months post-vaccination in our cohort of healthcare workers who have received the two-dose regimens of BNT162b2. In this cohort of vaccinated subjects, 43% (*n* = 25/58) of the participants no longer exhibit NAbs activity 180 days after the administration of the first dose of BNT162b2.

## 1. Introduction

The detection of antibodies directed against the spike protein (S protein), the receptor-binding domain (RBD), or the nucleocapsid (N) in the serum or plasma of convalescent patients allows monitoring of the development of adaptive immunity after SARS-CoV-2 infection or vaccination [1,2,3]. Following infection or vaccination, numerous antibodies are produced targeting different epitopes of the virus or the spike protein. Nevertheless, not all these antibodies are able to efficiently neutralize the virus since they can bind an epitope which is not essential for the virus entry into the cell. Therefore, the detection of neutralizing antibodies (NAbs) is of particular importance because these are the antibodies that can prevent the binding of the RBD of the S protein to the angiotensin-converting enzyme 2 (ACE2) receptor present at the surface of human cells, preventing the entry of the virus into the host cells [4].

In their recent study, Montesinos et al. presented serological data obtained from a series of six binding assays and compared their clinical performances against a plaque reduction neutralization test (PRNT) at a 1/80 titer [5]. They concluded that “VIDAS IgG and Euroimmun QuantiVac IgG showed a better ability to detect NAbs” with an area under the ROC curve of 0.96 (95% CI: 0.93–0.98) and 0.95 (95% CI: 0.92–0.98), respectively. According to the authors, the Roche assays, i.e., targeting the nucleocapsid or the spike protein, were less able to detect neutralizing antibodies (NAbs) with AUC of 0.87 (95% CI: 0.81–0.92) and 0.88 (95% CI: 0.82–0.93), respectively. They also reported a significant decrease in NAbs in 16.9% (11/65) subjects at the 6 months follow-up. Using their data, 33.3% (21/63) of the subjects lost their positivity with the VIDAS IgG and 27.7% (18/65) with the Euroimmun QuantiVac IgG. Having these data in mind, it is difficult to definitely conclude that these tests showed a better ability to detect NAbs than the other tests evaluated in this study [5]. In addition, our group and others previously showed that the positivity threshold reported in the instructions for using the Roche Elecsys^®^ anti-SARS-CoV-2 spike is not a threshold for correlating with neutralization [6,7,8,9,10,11]. It was demonstrated that higher antibody titers are needed to correlate with seroneutralization although this is subject to the diversity of antibody response among individuals [7,11]. This has also been observed in vaccinated subjects [6].

## 2. Materials and Methods

To complement the data from Montesinos et al., we would like to share our experience on a similar cohort of SARS-CoV-2 infected individuals and report the evolution of the positivity rate of the different serological tests over time versus a validated pseudovirus neutralization test (pVNT). In a cohort of 62 RT-PCR confirmed COVID-19 patients, 114 samples were collected from day 1 to day 296 after symptom onset; the average age of the population was 55.66 years (range 24–95), the gender was 50% male and 50% female, and the hospitalization rate was 25.80%.

In this cohort, we assessed six commercial binding assays, namely: the Roche (Basel, Switzerland) nucleocapsid (NCP) total antibody assay (positivity cut-off = 1.0 cut-off index (COI)), the Roche RBD total antibody assay (positivity cut-off = 0.8 U/mL), the DiaSorin (Saluggia, Italy) S1/S2 IgG assay (positivity cut-off = 15 AU/mL), the Ortho (Raritan, NJ, USA) S1 IgG assay (positivity cut-off = 1.0 S/V (sample signal/threshold value)), the Ortho S1 total antibody assay (positivity cut-off = 1.0 S/V), and the Phadia (Portage, MI, USA) S1 IgG assay (positivity cut-off = 0.7 U/L) and two neutralizing techniques, a surrogate virus neutralization test (sVNT) (positivity cut-off = 10 AU/mL) and a pVNT (positivity cut-off = 1/20 dilution factor of serum sample), as described previously [8,12,13,14]. All of these samples were collected before the SARS-CoV-2 vaccination campaign, which started in Belgium in January 2021. Descriptive statistics were used to analyze the data. ROC curves were performed to define cut-off values with the best specificity and sensitivity for each serological test. Sensitivity, specificity, positive and negative predictive values, as well as the accuracy, were calculated for the cut-offs provided by the manufacturer as well as for the adapted cut-offs obtained with the ROC curves as determined by Youden’s index. A simple linear regression and Pearson correlations were computed to assess the potential association between NAb and antibody titers. Inter-rater agreements were also determined. Data analysis was performed using GraphPad Prism^®^ software (version 9.1.0, San Diego, CA, USA) and MedCalc^®^ Software Ltd. (Diagnostic test evaluation calculator, Version 20.014. Available online: https://www.medcalc.org/calc/diagnostic_test.php accessed 12 October 2021).

## 3. Results and Discussion

Using the cut-off provided by the manufacturers, the sensitivity did not statistically differ between the different serological assays investigated in this study (Table 1). The specificity was below 50.0% for all assays meaning that these assays generated many false-positive results for the detection of NAbs. The accuracy for detecting NAbs was, in general, below 90% and stayed around 80% for all assays. The use of adapted cut-offs does not permit to improvement in the accuracy of these serological assays. It means that we needed to deal with a loss of sensitivity or specificity in order to increase to other parameters (i.e., specificity or sensitivity and accuracy). However, there was no real possibility to improve the capacity of these tests to detect NAbs by adapting the cut-offs (Table 1 and Table 2).

It is also interesting to analyze in more detail the samples that showed discordant results between pVNT and serological assays (Figure 1 and Table 2). In general, serological assays targeting total antibodies, i.e., Roche RBD, Roche NCP, and Ortho S1 total antibodies generated less false-negative results. This meant that also measuring both IgA and IgM permitted to better evaluate the neutralizing capacity of the serum than just targeting IgG. In general, the rate of false-negative was around 3% for assays targeting total antibodies while it was around 10% for assays specific towards IgG except for the Phadia S1 IgG, which showed a false negative’s rate of 2.2%. Nevertheless, this is at the detriment of a higher false-positive rate, which may be up to 84.0% for the Roche NCP total antibody assay. On the other hand, assays specific for IgG do not show very good performance either since their false positive rate was around 50 to 80%. These results were also confirmed by the ROC curves analyses (Figure 2). Therefore, binding immunoassays which showed the best agreement with NAbs as determined by a pVNT method are the Roche total RBD, Roche total NCP, and the Ortho S1 total antibodies. Nevertheless, all of these binding immunoassays demonstrated poor correlation with results obtained in pVNT (Appendix A). These results are confirmed by the ROC curves analyses (Figure 2). 

It is interesting to compare the results obtained with the sVNT which targets NAbs with those obtained with the pVNT technique (Figure 1). This test showed the best correlation with the pVNT technique, i.e., 32.0% of false-positive and 2.2% of false negative. The majority of the false-positive results (6/8) presented in Table 2 are close to the cut-off of the sVNT technique ranging from 10.05 AU/mL to 16.24 AU/mL. Based on the ROC curve analysis (Figure 2), the adapted cut-off of the sVNT increases from 10 AU/mL to 16.6 AU/mL. This new cut-off increases the specificity from 65.2% to 92.0% without significantly impacting the other computed parameters. This differs from other serological tests where an adapted cut-off increases specificity while negatively impacting other parameters. The correlation between the pVNT results and the sVNT results is better than the other tests, but the data remains very heterogeneous (Appendix A).

This study also permits the evaluation of the dynamic of NAbs production after symptoms onset in patients suffering from COVID-19. For a part of the population, a longitudinal follow-up with at least two samples for the same patient was performed (Figure 3A).

From day 14 to day 291, more than 75% of the samples were positive for NAbs (*n* = 87/110, 79.1%). Interestingly, 6 months post symptoms onset, the majority of the samples (*N* = 44/52, 84.6%) were still positive for NAbs.

This is in sharp contrast with the results we obtained 6 months post-vaccination in our cohort of healthcare workers who received the two-dose regimens of BNT162b2 [6].

In this cohort of vaccinated subjects, 43% (*n* = 25/58) of the participants no longer exhibit NAbs activity 180 days after the administration of the first dose of BNT162b2 [6]. This is a very interesting observation since even those who were seropositive at baseline, i.e., a documented previous infection to SARS-CoV-2, seemed to lose their neutralizing capacity (*n* = 7/18, 39%) [6]. (Figure 3B) The comparison between the results obtained in COVID-19 patients and those obtained in vaccinated individuals with or without previous COVID-19 infection is presented in Figure 4.

C. Jeewandara et al. also pointed out a relationship between the development of NAbs and the time of exposure to the virus. In their study, patients with prolonged exposure to SARS-CoV-2 had a higher NAbs titer than patients who cleared the virus earlier [15]. Such differential response could be explained by a longer exposure to the antigen(s) during the course of COVID-19 disease compared to vaccination, where it is supposed that the antigen is rapidly cleared from the body. These results deserve further investigations to better understand the difference in the dynamic of antibody production after COVID-19 disease and vaccination.

## Figures and Tables

**Figure 1 viruses-13-02329-f001:**
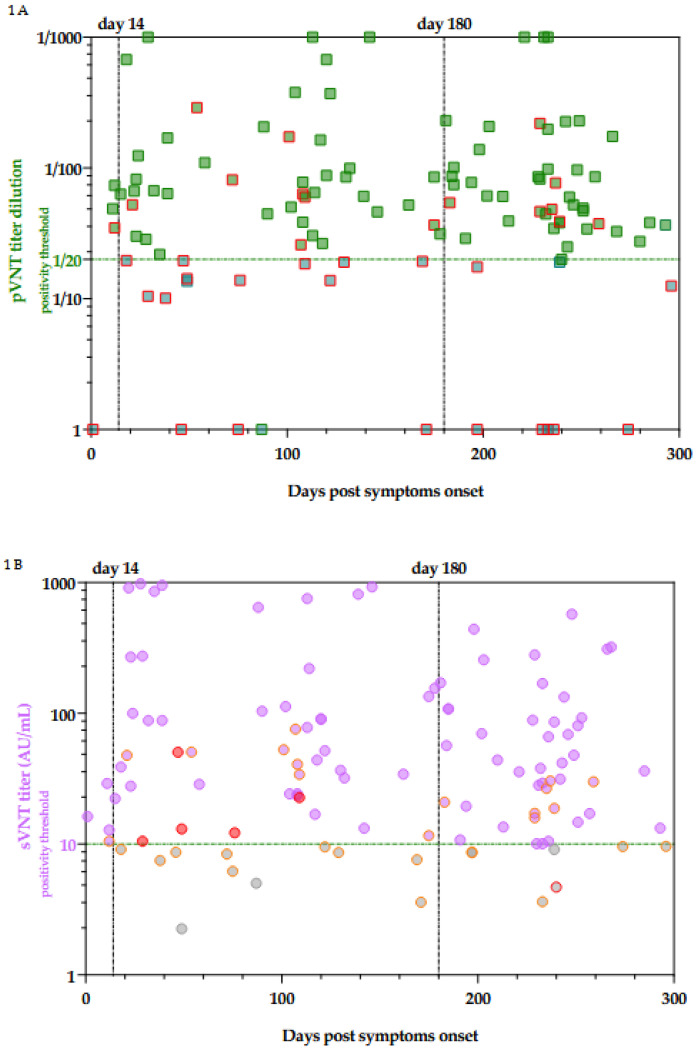
Evolution of NAbs titer in pVNT (**A**) and sVNT (**B**) as a function of days post symptoms onset. Green squared and purple dots are concordant results. Squares framed in red are those for which there is a discordant result between pVNT result and at least one of the six non-neutralizing immunoassays. Dots circled in orange are those for which there is a discordant result between sVNT and at least one of the six non-neutralizing immunoassays. Dots circled in red are those for which only sVNT was discordant. Red dots are those for which sVNT is discordant from pVNT. The details of these discrepancies are reported in Table 2.

**Figure 2 viruses-13-02329-f002:**
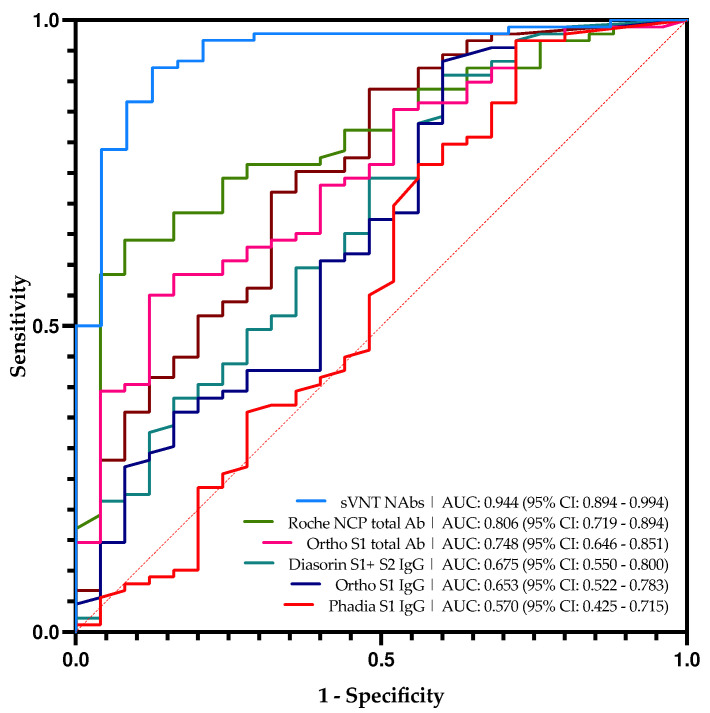
Roc curve analyses of all serological tests in this study and sVNT technique in comparison with pVNT based on the manufacturer’s cut-offs.

**Figure 3 viruses-13-02329-f003:**
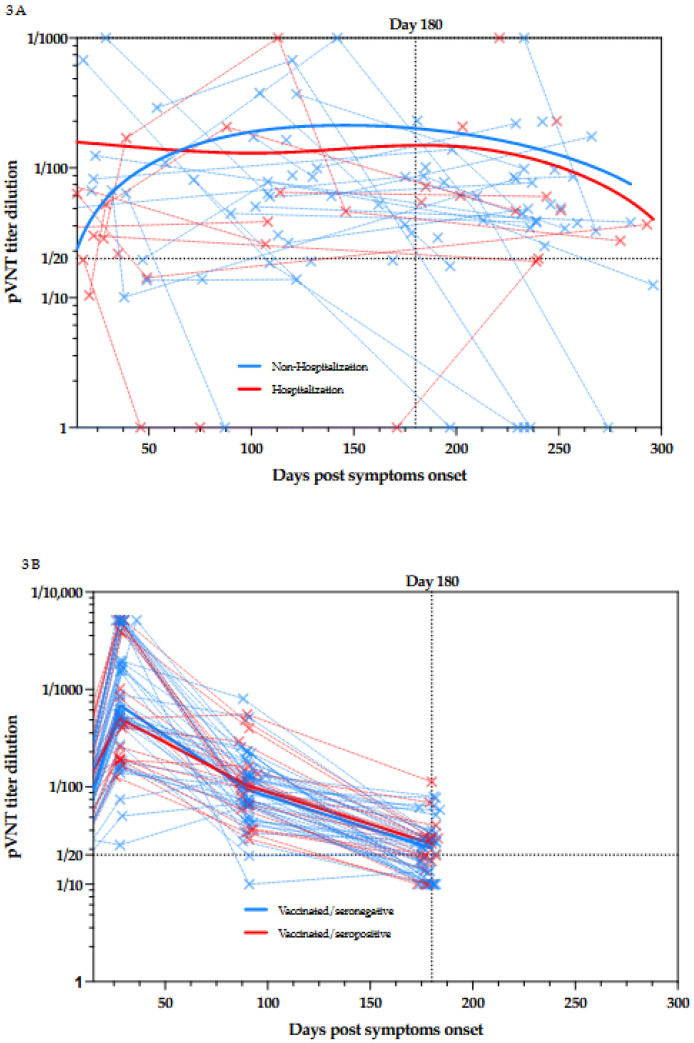
(**A**) Kinetics of NAbs since the symptom onset in non-hospitalized and hospitalized patients. Blue plain lines represent non-hospitalized patients and red plain lines represent hospitalized patients. Samples obtained at different times for the same patient are connected by dotted lines. (**B**) Kinetics of NAbs since the day of first dose vaccination in seronegative and seropositive individuals. Blue plain lines represent seronegative individuals and red plain lines represent seronegative individuals. Samples obtained at different times for the same individual are connected by dotted lines.

**Figure 4 viruses-13-02329-f004:**
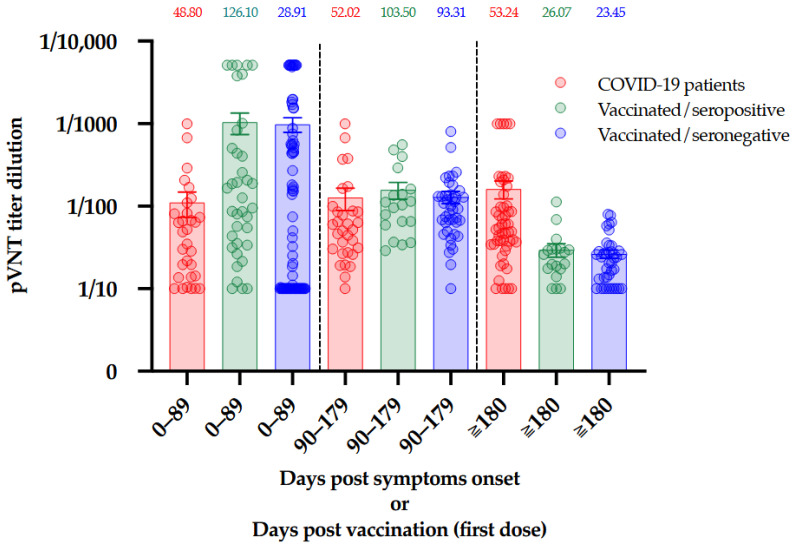
Comparison of NAbs titers between COVID-19 patients, seropositive vaccinated individuals, and seronegative vaccinated individuals at three different timeframes. The median is presented on top of each timeframe for each population. Bars represent the mean and standard error or mean.

**Table 1 viruses-13-02329-t001:** Summary table of the analytical sensitivity, specificity, positive predictive value, negative predictive value, and accuracy of the six serological assays investigated in this study and sVNT technique in comparison with the pVNT. Results obtained with the adapted cut-offs are presented in italics.

Serological Assay	Cut-Off Definition	Sensitivity (95% CI)	Specificity (95% CI)	PPV (95% CI)	NPV (95% CI)	Accuracy (95%CI)
**Roche RBD total antibody assay**	Manufacturer: 0.8 U/mL	97.8% (92.2–99.7%)	33.3% (15.6–55.3%)	84.6% (80.5–88.0%)	80.0% (47.6–94.6%)	84.2% (76.2–90.4%)
	*Adapted:* *5.9 U/mL*	*88.8%* *(80.3–94.5%)*	*52.0%* *(31.3–72.2%)*	*88.1%* *(83.0–91.8%)*	*53.6%* *(36.6–69.8%)*	*81.4%* *(73.0–88.1%)*
**Roche NCP total antibody assay**	Manufacturer: 1.0 COI	97.8% (92.2–99.7%)	12.5% (2.7–32.4%)	80.7% (78.2–83.0%)	60.0% (21.0–89.5%)	79.8% (71.3–86.8%)
	*Adapted:* *37.7 COI*	*61.8%* *(50.9–71.9%)*	*84.0%* *(63.9–95.5%)*	*93.9%* *(86.11–97.5%)*	*35.5%* *(28.6–42.9%)*	*66.2%* *(56.8–74.8%)*
**DiaSorin S1/S2 IgG assay**	Manufacturer: 15.0 AU/mL	87.6% (79.0–93.7%)	44.0% (24.4–65.1%)	86.2% (81.4–89.9%)	47.1% (30.5–64.4%)	78.9% (70.3–86.0%)
	*Adapted:* *11.4 AU/mL*	*91.0%* *(83.1–96.0%)*	*40.0%* *(21.1–61.3%)*	*85.9%* *(81.4–89.4%)*	*52.6%* *(32.9–71.6%)*	*80.8%* *(72.4–87.6%)*
**Ortho S1 IgG assay**	Manufacturer: 1.0 S/V	90.0% (81.9–95.3%)	41.7% (22.1–63.4%)	85.3% (80.4–89.1%)	52.6% (33.8–70.8%)	79.8% (71.3–86.8%)
	*Adapted:* *0.3 S/V*	*93.3%* *(85.9–97.5%)*	*40.0%* *(21.1–61.3%)*	*86.1%* *(81.8–89.6%)*	*59.7%* *(37.4–78.6%)*	*82.6%* *(74.4–89.1%)*
**Ortho S1 total antibody assay**	Manufacturer: 1.0 S/V	98.9% (94.0–100.0%)	16.7% (4.7–37.4%)	81.7% (78.8–84.2%)	80.0% (73.2–88.2%)	81.6% (73.2–88.2%)
	*Adapted:* *165.0 S/V*	*53.9%* *(43.0–64.6%)*	*84.0%* *(63.9–95.5%)*	*93.1%* *(84.3–97.1%)*	*31.3%* *(25.6–37.7%)*	*59.9%* *(50.3–69.0%)*
**Phadia S1 IgG assay**	Manufacturer: 0.7 U/L	97.1% (92.1–99.7%)	16.0% (4.5–36.1%)	82.3% (79.6–84.7%)	64.0% (25.7–90.2%)	81.4% (73.0–88.1%)
	*Adapted:* *2.5 U/mL*	*96.6%* *(90.5–99.3%)*	*28.0%* *(12.1–49.4%)*	*84.3%* *(80.7–87.3%)*	*67.5%* *(36.6–88.2%)*	*82.9%* *(74.7–89.3%)*
**sVNT**	Manufacturer: 10 AU/mL	97.8% (92.3–99.7%)	65.2% (42.7–83.6%)	91.8% (86.5–95.2%)	88.1% (64.6–96.8%)	91.3% (84.5–95.7%)
	*Adapted:* *16.6 AU/mL*	*97.7%* *(92.1–99.7%)*	*92.0%* *(74.0–99.0%)*	*98.0%* *(92.8–99.5%)*	*91.1%* *(72.1–97.6%)*	*96.6%* *(91.4–99.1%)*

**Table 2 viruses-13-02329-t002:** Summary of discrepancies between pVNT versus serological assays and sVNT in samples from patients with COVID-19. The total number of pVNT negative and positive samples were 25 and 89, respectively. Discordant results between pVNT and serological tests are highlighted in orange.

Days Since Symptoms Onset	pVNT Dilution Factor (>20 = POS)	sVNT AU/mL (>10 = POS)	Roche RBD Total Ab U/mL (≥0.8 = POS)	Roche NCP Total Ab U/mL (≥1.0 = POS)	Ortho S1 Total Ab S/V (≥1.0 = POS)	Ortho S1 IgG S/V (≥1.0 = POS)	Diasorin S1 + S2 IgG AU/mL (≥15 = POS)	Phadia S1 IgG U/mL (>0.7 = POS)
	**False positive** | **PVNT Negative**/**Serological Positive**							
**1**	10.00	16.24	1.13	1.50	1.83	0.24	19.10	1.20
**18**	19.60	9.12	0.4	0.20	1.32	0.05	27.60	0.70
**29**	10.46	10.50	174.00	33.20	255.00	11.50	78.80	73.00
**38**	10.09	7.49	0.40	5.06	0.31	0.01	3.80	0.70
**46**	10.00	8.66	0.40	24.50	1.87	0.15	7.00	5.60
**47**	19.54	50.36	127.00	32.30	138.00	9.61	58.00	28.00
**49**	14.24	13.04	57.30	19.10	60.30	9.64	47.00	24.00
**75**	10.00	6.19	322.00	37.20	271.00	12.50	86.80	250.00
**76**	13.86	12.20	153.00	22.60	126.00	12.30	73.10	58.00
**109**	18.51	22.69	2219.00	114.00	552.00	19.00	311.00	186.00
**122**	13.82	9.53	0.61	20.80	4.57	0.20	11.00	9.00
**129**	19.06	8.61	2.44	4.75	47.90	2.75	6.30	7.80
**169**	19.34	7.59	3.36	15.00	18.90	0.19	3.80	1.90
**171**	10.00	3.59	42.30	15.50	109.00	7.37	62.30	36.00
**197**	10.00	8.69	476.00	48.30	136.00	17.30	141.00	790.00
**197**	17.51	8.59	41.60	26.40	164.00	7.20	32.10	31.00
**230**	10.00	10.05	4.80	2.38	12.10	5.05	11.10	11.00
**233**	10.00	3.63	0.75	14.34	0.54	0.53	11.53	8.71
**236**	10.00	10.50	0.40	11.30	7.77	0.15	9.60	9.90
**239**	19.05	9.09	31.90	3.20	60.10	9.93	48.60	34.00
**274**	10.00	9.58	191.00	9.57	120.00	14.10	132.00	1950.00
**296**	12.52	9.61	5.33	9.20	12.70	5.70	43.80	18.00
**False Positive**	**NA**	**8/25** **(32.0%)**	**16/25** **(64.0%)**	**21/25** **(84.0%)**	**20/25** **(80.0%)**	**14/25** **(56.0%)**	**14/25** **(56.0%)**	**20/25** **(80.0%)**
	**False Negative** | **PVNT Positive**/**Serological Negative**							
**12**	34.87	10.52	27.50	3.95	39.30	2.61	14.20	5.80
**21**	52.36	47.69	11.70	57.30	22.60	6.08	12.70	19.00
**54**	289.70	50.55	2.70	3.36	4.81	0.07	20.90	0.70
**72**	81.25	8.40	4.52	13.60	10.10	0.24	7.50	3.80
**101**	173.09	52.61	1.61	16.00	2.41	0.86	7.00	3.00
**107**	25.90	75.66	0.40	3.94	4.36	0.20	8.12	14.00
**108**	63.18	40.70	18.40	27.80	33.80	0.57	7.10	3.20
**175**	36.67	11.60	0.40	0.07	0.06	0.01	3.80	0.70
**183**	57.18	20.90	0.81	0.46	1.34	0.33	6.00	10.00
**229**	219.43	17.21	65.90	90.40	1.55	0.17	46.80	27.00
**229**	46.52	15.90	3.60	28.70	10.90	3.60	9.80	11.00
**235**	48.41	26.68	7.76	95.90	13.30	3.55	11.70	11.00
**237**	76.84	30.44	12.10	44.70	23.30	4.92	10.00	12.00
**240**	20.00	4.69	142.00	37.60	116.00	13.70	103.00	204.00
**259**	37.48	29.97	1.26	5.09	2.38	0.14	24.40	1.00
**False negative**	**NA**	**2/89** **(2.2%)**	**2/89** **(2.2%)**	**2/89** **(2.2%)**	**1/89** **(1.1%)**	**9/89** **(10.1%)**	**11/89** **(12.3%)**	**2/89** **(2.2%)**
**Inter rater agreement (95% CI)**	**NA**	**0.72** **(0.56–0.88)**	**0.38** **(0.17–0.59)**	**0.14** **(0.0–0.31)**	**0.21** **(0.02–0.40)**	**0.33** **(0.12–0.54)**	**0.33** **(0.12–0.54)**	**0.19** **(0.0–0.38**

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
