# Peer review of "Dynamics of Neutralizing Antibody Responses Following Natural SARS-CoV-2 Infection and Correlation with Commercial Serologic Tests. A Reappraisal and Indirect Comparison with Vaccinated Subjects"

_viruses, 2021, doi:10.3390/v13112329_

Round 1
Reviewer 1 Report
Nice paper. No criticism to be highlighted
Reviewer 2 Report
In a cohort of 75 PCR-positive convalescent individuals, Gillot et al measure SARS-CoV-2 binding antibodies using 6 commercial serological tests, neutralising antibodies (Nabs) using the surrogate virus neutralising test (SVNT) and compare the results to Nabs measured via a pseudovirus neutralising test (PVNT). The authors suggest new cut-offs for the serological tests and SVNT to more accurately predict whether a sample contains Nabs. The authors show that a large proportion of convalescent subjects maintain Nabs at 6 months post-symptom onset, and indirectly compare it to one of their previous studies showing that Nabs in vaccinated individuals wane more significantly. There are a few concerns in this manuscript that need to be addressed and the longitudinal data could be presented more clearly.
- The manuscript starts rather abruptly with the description of the study by Montesinos et al. The authors should write a few sentences describing the different serological tests for COVID-19 (total Ab vs IgG, different antigen targets), and describe how only a subset of these binding antibodies would be neutralising antibodies.
- Line 17 – PRNT should be plaque reduction neutralisation test
- Line 53 – the authors state that spearman correlations were computed but pearson correlations are presented in Fig S1.
- Line 63-67 – this sentence is rather long and confusing.
- Figure 1 is quite convoluted and it is hard to draw any meaningful trends from the data in its current format. I suggest separating the pVNT and sVNT data into two separate graphs.
- Lines 112-114: How many of the samples in Figure 1 were longitudinal samples from the same donors? Longitudinal samples could perhaps be plotted on a separate curve with connecting lines or a trendline to more accurately display the waning of neutralising antibodies over time. Do the 52 samples at 6 months post-symptom onset have matched samples from earlier timepoints?
- Figure 2 – the legend should specify whether this data was calculated using the manufacturer’s or adapted cut-offs.
- Table 2 – I don’t see the significance of the “days since symptoms onset” column. The false positive/negative rates don’t seem to increase or decrease with days post-symptom onset. Maybe include a column with participant ID as well?
- Lines 116-120 – was the same pseudovirus neutralising test used for the cohort of BNT162b2-vaccinated subjects? It is odd that the percentage of convalescent subjects at 6 months post-symptom onset still positive for Nabs is higher than the percentage of vaccinated convalescent subjects at 6 months post-vaccine positive for Nabs. Did a large proportion of the convalescent cohort described in this manuscript have severe COVID-19?
- The title of the manuscript should not include “comparison with vaccinated subjects” since no data from vaccinated subjects are shown in this manuscript.
- The authors should include a table describing the characteristics of the cohort, including age, gender and disease severity
Reviewer 3 Report
Reviewer #1:
Coronavirus disease 2019 (COVID-19) has emerged as a new world pandemic, infecting millions of people with a substantial mortality.
There is significant interest in understanding of neutralizing antibody responses to natural SARS-CoV-2 infection and also vaccination. Commercial serologic tests are critical to know the levels of immunity in the people after infections.
Therefore, these information in the review could be help in understand this time immunity to natural infection.
In this communication, by Douxfils et al titled "Dynamics of neutralizing antibody responses following natural SARS-CoV-2 infection and correlation with commercial serological test. A reappraisal and indirect comparison with vaccinated subjects". The authors performed an analysis of SARS-CoV-2 infected individuals and the evolution of the positivity rate of the different serological test over time versus a validated pseudovirus neutralization test (pVNT).
The positives results obtain in relation to 6 moths post symptoms onset was interesting versus the results found 6 moths’ post-vaccination.
There are minimal concerns that to be addressed.
This manuscript is well written and sites key findings in the field, therefore it will be helpful for investigators entering into coronavirus/COVID-19 research. The study would benefit the section on general aspects concern to dynamic of antibody production after COVID-19 disease or post vaccination. Comments to improve the clarity of the manuscript are provided below.
Comments for the authors' consideration:
- Double check small typing errors in all manuscript.
- Please add a figure with a correlation of NAbs titer in natural infection versus vaccination).
Round 2
Reviewer 2 Report
I thank the authors for addressing my comments. Apologies, my original comment that said "the sentence on lines 63-67 was long and confusing" was actually referring to the sentence beginning with "The use of adapted cut-offs does not permit to improve the accuracy..." on lines 87-91. It would be good if the authors could rephrase this sentence.
